# Using Population-Based Structures to Actively Monitor AEFIs during a Mass Immunization Campaign—A Case of Measles–Rubella and Polio Vaccines

**DOI:** 10.3390/vaccines9111293

**Published:** 2021-11-08

**Authors:** Dan Kajungu, Victoria Nambasa, Michael Muhoozi, Joan Tusabe, Beate Kampmann, Jim Todd

**Affiliations:** 1Makerere University Centre for Health and Population Research (MUCHAP), Makerere University, Kampala P.O. Box 7062, Uganda; muhoozimusi@gmail.com (M.M.); jtusab@gmail.com (J.T.); 2Department of Global Health, Stellenbosch University, Stellenbosch 7602, South Africa; 3National Pharmacovigilance Centre, National Drug Authority (NDA), Kampala P.O. Box 23096, Uganda; vnambasa@nda.or.ug; 4The Vaccine Centre, London School of Hygiene and Tropical Medicine, Keppel Street, London WC1E 7HT, UK; Beate.Kampmann@lshtm.ac.uk; 5Department of Population Health, London School of Hygiene and Tropical Medicine, Keppel Street, London WC1E 7HT, UK; Jim.Todd@lshtm.ac.uk

**Keywords:** active monitoring, pharmacovigilance, AEFIs, mass vaccination campaigns

## Abstract

Active vaccine pharmacovigilance complements the standard passive or spontaneous surveillance system, which suffers from low reporting rates. This study aimed at utilizing population-based structures to report and profile adverse events following immunization (AEFI) with the measles and rubella vaccine (MR), or MR in combination with the bivalent oral polio vaccine (bOPV 1&3) (MR & bOPV), during mass vaccination in Uganda. Caretakers of children at home (less than 5 years) and schoolgoing children were followed up on and encouraged to report any AEFIs on day one, 2–3 days, 10 days, and 14 days after vaccination at school by their teachers and at-home, community-based village health teams. Out of 9798 children followed up on, 382 (3.9%) reported at least one AEFI, and in total, 517 AEFIs were reported. For MR, high temperature (21%), general feeling of weakness (19.3%), and headache (13%) were the most reported AEFIs, though there were variations on the days when they were reported. For the combination dose of MR & bOPV, high temperature (44%), rash (17%), general feeling of weakness (13%), and diarrhoea (8%) were the most common adverse events following immunization reported by caretakers. All 382 children cleared the AEFIs within 2 days, with 343 (90%) children reporting mild or moderate AEFIs and only 39 (10%) reporting severe AEFIs. The reported AEFIs are known and are mentioned in the vaccine leaflets with similar severity classification. Rates of AEFIs differed with the number of days after receiving the immunization. Conclusion: Active surveillance for AEFIs provides additional important information to national vaccine regulatory bodies. It reassures the public that vaccines are safe and that their safety is being taken seriously in Uganda, which would improve vaccine acceptability and confidence in the health system. Piggybacking on existing structures such as village health team members (for children at home) and teachers (for schoolgoing children) facilitates reaching vaccine recipients and increases reporting rates. Therefore, studies using active reporting of AEFIs should be conducted at regular intervals to report the overall incidence of AEs and to monitor trends and changes.

## 1. Introduction

Vaccination is one of the most cost-effective interventions for global health, which has led to protection against transmissible infections in recent decades, leading to an expressive impact on child morbidity and mortality [1,2,3,4]. Despite these significant contributions, the general public has concerns about potential vaccine-associated risks, including adverse events following immunization, which can undermine such successes. A “vaccine adverse event”, also referred to as an “adverse event following immunization” (AEFI), is an adverse health event or health problem that occurs following or during administration of a vaccine. Adverse events are temporally associated events; they might be caused by a vaccine, or they might be coincidental and not related to vaccination [5,6].

National immunization programs aim at increasing uptake of vaccines in the population to mitigate the impact of vaccine-preventable diseases (VPDs). Mass immunization as a strategy for increased uptake leads to large numbers of vaccine exposures, which means that more events occur among the exposed population irrespective of their severity. In addition, concerns related to vaccine safety, which may be baseless or legitimate, tend to emerge. These concerns may be falsely attributed to the memorable immunization event or may be real vaccine-induced reactions [6]. It is therefore important to differentiate these types of concerns by implementing appropriate surveillance systems for AEFIs, with timely and thorough scientific assessment.

In mass immunization campaigns where large numbers of doses are administered in a short time, AEFIs tend to be more noticeable to the public and healthcare personnel [7]. Regardless of the specific cause, AEFIs may cause unnecessary and unjustified suspicion towards vaccines in the public because vaccines are administered to children and individuals who are not sick. Therefore, explaining the benefits as well as the risks associated with vaccines and engaging caretakers and recipients mitigates the negative attitude towards further immunization uptake. That way, the risk of susceptibility to VPDs that are disabling and life-threatening is mitigated.

To sustain public confidence, compliance, and acceptance of immunization programs, it is important to have active safety surveillance to find AEFIs during mass immunization campaigns and to report these events openly and transparently. According to the World Health Organization, all adverse events that are of concern to the caregiver and have associated costs and effects should be reported regardless of severity [8,9]. Active surveillance systems for vaccine safety monitoring form an important part of surveillance systems with established capacity and infrastructure [10]. The identification and evaluation of longitudinal demographic, health, and vaccination program information and data collection systems are invaluable in safety signal generation. The value of such systems can help in the analysis and replication of already-known adverse event associations such as febrile seizures following measles vaccination [10].

Low- and middle-income countries (LMICs) have relatively weaker systems and capacity for vaccine evaluation and safety monitoring post-licensure and therefore often depend on similar data from high resource countries [10]. Uganda, like the majority of LMICs, largely depends on passive surveillance systems to generate and detect new safety signals, but these passive systems often suffer from low reporting. There is, therefore, a need to complement passive surveillance with well-integrated strategies for active reporting of AEFIs that recognize and manage risks and increase benefits in a cost-effective manner. Such initiatives, if well maintained, can contribute to better profiling and quantifying of the risk associations between an administrated vaccine and a potential AEFI.

In October 2019, Uganda carried out a national mass vaccination campaign against measles, rubella, and polio using the measles and rubella (MR) live vaccine, attenuated and the bivalent oral polio vaccine (bOPV 1&3). The campaign was implemented in response to measles and rubella outbreaks in which suspected measles cases were reported, patients were admitted with symptoms of measles–rubella disease, and measles–rubella related deaths were recorded in the three years’ period of the outbreak. The vaccination campaign engaged schools, vaccination posts, village health team (VHT) members, and local councils (LC1) in addition to local government structures at the parish, subcounty, health subdistrict, district levels.

The aim of this study was to profile the patterns of AEFIs occurring among children vaccinated during the national measles–rubella–polio (MR & bOPV) mass vaccination campaign through active vaccine safety monitoring strategies in Iganga district, Uganda.

## 2. Materials and Methods

A prospective active surveillance system followed up with children either in the communities or at school for 14 days during the MR & bOPV national mass immunization campaign in Iganga district. Enrolment commenced on 18 October 2019, and the follow-up was completed on 31 October 2019. This was a complimentary system alongside the passive reporting system adopted by the National Drug Authority (NDA) of Uganda to collect AEFIs from the immunization program. It was implemented in schools and communities in a population-based cohort.

### 2.1. Setting and Population

The surveillance was nested within a population-based cohort of Iganga Mayuge health and demographic surveillance site (IMHDSS) in Iganga and Mayuge districts [11]. The demographic surveillance area (DSA) covered 65 villages with total population of over 90,000 residents from 18,634 households, with 51% females. There were 23 health facilities within and at the borders of the DSA, including 1 public general hospital, 15 level II health centres (HC) (eleven public, three PNFP and one private-for-profit), 6 level III HCs (five public and one PNFP), 1 level IV HC (public), 2 private clinics, and over 150 drug shops. There were over 100 primary school and preschool setups. The IMHDSS cohort uniquely identified all members at the household and individual level, making follow-up possible whenever necessary.

### 2.2. Study Procedure

All children aged less than 15 years were vaccinated during the national MR & bOPV immunization campaign. The MR vaccine was administered to children ≥59 months and below 15 years whether previously immunized or not, while those below 59 months of age also received bOPV. They were followed up with by teachers at school and community mobilizers known as village health teams (VHTs) in communities. The teachers and VHTs were given basic training on vaccine safety monitoring and vigilance, research ethics, and reporting. All immunized children and the caretakers (parents or legally acceptable representative) were oriented to study procedure. All participants were monitored at the immunization site in the waiting area for 30 min following their vaccinations for the occurrence of any events. Parents of children and schoolgoing children were provided with leaflets in both English and Lusoga (local language) that had information on vaccination and possible AEFIs.

The teachers monitored children immediately after immunization and in the following days while at school. The trained VHTs either visited households or contacted the caretakers on day one, 2–3 days, 10 days, and 14 days after immunization. Teachers directly asked schoolgoing children, while VHTs asked the caretakers if the children had experienced any adverse event and if they suspected this could have been as a result of the vaccine received during the mass vaccination campaign. A research supervisor worked with the teams to observe and manage any AEFI or refer children to the nearest health facility for further management when it was considered serious. The choice of days for follow-up was informed by literature. Studies on AEFIs and adverse vaccine reactions have shown that such events are approximated to happen at 6 to 14 days after vaccination with measles–rubella vaccines [12,13]. The telephone contact details of the supervisors were availed to all participants. Data were collected using an easy to complete paper-based case report tool developed in collaboration the National Pharmacovigilance Centre (NPC) to capture AEFIs, including the predefined adverse events of interest. All reports collected through this active follow-up were shared with the National Pharmacovigilance Centre (NPC) of the National Drug Authority (NDA) and the Uganda Expanded Program on Immunization (UNEPI) at the Ministry of Health for further analysis or assessment and were added to the national line listing.

### 2.3. Data Processing and Analysis

Data were double entered in EPIDATA version 3.02, consistency checks were performed, and all queries were resolved before locking the database and transferring the data to STATA version 15 for analysis. Because, in some instances, multiple AEFIs were reported by a single child, the STATA *mrtab* command, which tabulates multiple responses that are held as a set of indicator variables or as a set of polytomous response variables, was applied because of the data-storage mode, that is, the indicator mode. The results were described using tables and bar graphs. The proportion of AEFIs was computed for all studied children who received the vaccine during the mass immunization exercise.

## 3. Results

A total of 10,174 participants were vaccinated in the selected villages and schools in the IMHDSS. Of these, 256 (2.5%) were lost to follow up, and 120 (1.2%) declined to participate in the study. Finally, 9798 (96.3%) children were considered for analysis, with 4206 (42.9%) children vaccinated from villages and 5592 (57.1%) from schools. Among those who reported at least one AEFI, 163 (42.7%) had been vaccinated in the community (and the caretakers/parents reported to the VHT), while 219 (57.3%) had been vaccinated at schools, as shown in Figure 1.

### 3.1. Description of AEFI Characteristics

A total of 382 (3.9%) children experienced at least one AEFI, and a total of 517 AEFIs were recorded altogether. There was no difference in the proportion of children reporting any AEFI from schools or in the communities as in Figure 1. There were more females (57.1%) than males; the median age was 7.5 years. The majority of the children aged above five years were vaccinated at school (83%), while most of those under five were vaccinated in the communities (86.7%). More AEFIs were reported at school (305/517) than in the community. The time of AEFI occurrence indicated that 55% of the 302 AEFIs reported at school were reported on day one, and these contributed 73% of the total AEFIs reported on day one (165/225). More than half of the AEFIs reported in the community were recorded on the second day after immunization (56% or 120/215). Only 78 AEFIs were reported after 10 days, with most of them (43/78) reported from schools, as shown in Table 1.

### 3.2. Adverse Event following Immunization (AEFI) Profiles

The adverse events reported were similar to the AEFIs labelled by the manufacturer in the product label. High temperature was the most reported event for both the mono dose (MR) and the combination (MR & bOPV). For the MR and bOPV 1&3 combination, the most frequently reported AEFIs in descending order were as follows: high temperature, followed by rash, general feeling of weakness, diarrhoea, vomiting, loss of appetite, and others. On the other hand, the most frequently reported AEFIs in children who were immunized with MR alone were as follows: high temperature, general feeling of weakness, headache, loss of appetite, painful injection area, stomach pain, rash, diarrhoea, vomiting, severe allergic reactions, mumps, bruises, and seizures.

Considering individual vaccines, out of the 517 AEFIs reported, 287 (55.5%) were in children vaccinated with MR alone, as shown in Figure 2. Mumps were reported only in children receiving MR alone.

### 3.3. Patterns and Time of Occurrence

There were variations in the days when AEFIs were reported both at school and from the communities. The mono MR was administered to school-going children and to a few non-school-going children in the community who were out of school at the time of immunization. For children receiving MR alone, most AEFIs were reported on the day of immunization, while for the MR & bOPV combination, more AEFIs were reported two or three days after the immunization.

### 3.4. Measles–Rubella (MR) and AEFI Profile

The mono (MR) vaccine was administered to 6140 children ≥59 months and below 15 years whether previously immunized or not, and only 3.9% (240/6140) reported at least one AEFI. High temperature (21%), general feeling of weakness (19.3%), and headache (13%) were the most reported AEFIs, though there were variations on the days when they reported. High temperature was the most reported overall, but the reports of this were most frequent on the 10th day. More children reported loss of appetite (19%) on the 10th day. Most vaccine recipients reported painful injection site only one day after immunization, and numbers were very low on subsequent days. There were more reports of diarrhoea, vomiting, and seizures on day 14 compared to the earlier days (Figure 3).

### 3.5. MR and Polio Vaccine

The MR live attenuated vaccine and the bivalent oral polio vaccine (bOPV 1&3) were administered together to 3658 children aged less than five years, and 3.9% reported at least one AEFI. Active AEFI reporting was performed by caretakers to community-based VHTs or by teachers in kindergartens and/or nursery schools. Overall, high temperature (44%), rash (17%), general feeling of weakness (13%), and diarrhoea (8%) were the most common AEFIs reported by this group. There were more cases of rash reported on day ten compared to the earlier days, and fewer cases of diarrhoea reported on day one compared to day 2–3 and day 10, while loss of appetite was consistently reported on all days, accounting for 6–7% of all AEFIs reported. Events such as stomach pain, serious allergic reactions, painful injection site, headache, bruises, and seizures were reported only within three days after immunization (Figure 4).

### 3.6. Severity

The severity of an AEFI was classified as mild, moderate, or severe. The classification was dependent on the vaccine recipient or the caretaker’s judgement or opinion and was not assessed by a clinician or according to clinical definitions. Ninety-six percent of the vaccine recipients reported no AEFIs at all. Out of the 382 (3.9%) who reported at least one AEFI, 343 (90%) categorized them as mild or moderate, and only 39 (10%) reported a severe AEFI (Figure 5). All events cleared within a few days from onset.

## 4. Discussion

The national mass vaccination campaign reached 10,174 children with either MR or MR & bOPV in the Iganga Mayuge health and demographic surveillance site (IMHDSS) located in Iganga and Mayuge districts. Through active vaccine safety surveillance, an exercise that involved village health workers or schoolteachers, fewer than 4% of children reported any AEFI, with similar proportions in both groups of children vaccinated, at schools and in villages. Of the reported AEFIs, 90% were mild or moderate, and all were similar to those labelled for the product.

The rate of AEFIs in the present study was higher than the rate reported by the Ministry of Health (UNEPI) for the entire country (0.0005%) [14]. This is a reflection of the difference between active vaccine safety monitoring and passive surveillance, which relies on self-reporting of AEFIs to health facilities and later to the Ministry of Health. This comparison is in agreement with studies done in LMICs wherein passive vaccine pharmacovigilance showed low reporting rates [15,16]. The MR vaccine used in the immunization programs (prequalified by WHO) is a live-attenuated vaccine containing the Edmonston strain of measles and the RA 27/3 strain of rubella [17]. Prospective studies that used a similar product found incidence ranging from 13.7% to 20.8% [18,19]. However, one used a smaller sample size of 278 [18], and the other was conducted over a period of three years [19].

This population-wide active vaccine safety monitoring initiative during a mass immunization campaign setting reported and documented occurrences of AEFIs with MR or MR & bOPV in rural Uganda. This initiative reached more vaccine recipients, who were able to report more AEFIs in this campaign compared to similar campaigns recommended by the WHO for LMICs [20]. This has also been observed in active surveillance of AEFIs implemented elsewhere, with associated improved reporting rates, completeness of reports, clarity on temporal linking, and greater precision for estimates compared to passive reporting [16,17,18].

Generally, more cases were reported from schools than from the communities. More AEFIs on the first day were reported from schools than from the communities (whence slightly more were reported on the second day). The target group for the campaign had more children of schoolgoing age vaccinated [14]. The surveillance established at school was more likely to easily identify children who missed school on day one than VHTs. Schools can play an important role in ensuring vaccination activities are a success [21], including by reporting AEFIs. Vaccine pharmacovigilance strategies need to integrate schools in reporting efforts to complement community vigilance efforts.

The most commonly reported AEFI for both the mono dose (MR) and the combination (MR & bOPV) was high temperature. This is similar to the results of other longitudinal studies [4,18,22,23]. Rash, diarrhoea, and vomiting were frequently reported by children vaccinated with MR & bOPV. The occurrence of these events has been noticeable in studies monitoring OPV alone [23] and a combination of OPV and other antigens in similar settings [24].

For the measles rubella vaccine alone (administered on children aged 5–15 years), reports of high temperature were highest after two weeks (day 10) which is a delay and may not be related to the vaccine. More so, more cases of rash were reported on day ten for children that received the combination vaccine (MR & bOPV) compared to the earlier days. This could be attributed to delays in manifestation of AEFIs. It is also possible that such AEFIs identified later could have had no relationship with the vaccine.

There were few severe AEFIs, and no serious events were reported. An AEFI is considered serious if it results in death, is life-threatening, requires inpatient hospitalization or prolongation of existing hospitalization, results in persistent or significant disability/incapacity, or is a congenital anomaly/birth defect. There was no such scenario for the reported AEFIs. Only 39 (10%) events were reported as severe by the vaccine recipient or caretaker. Equivalently, in the passive surveillance, there were just three serious cases nationwide, which were documented and confirmed by EPI during the campaign [14]. The findings are consistent with the literature, which has indicated low occurrence of serious AEFIs. Elsewhere, in Zimbabwe and Switzerland (2016 and 2017), the prevalence of serious AEFI ranged between 11 and 19.4% [25,26,27]. However, these studies analysed spontaneous reports over a period of 20 years for different vaccines, which may not be an actual representation of AEFIs identified during active monitoring. In this study, severity did not equal seriousness, as it was not assessed by a clinician or according to the clinical definitions. This was the recipient’s or a caretaker’s judgement or opinion. All events cleared within a few days from onset. None of the severe events reported met the criteria for seriousness.

This study prospectively followed up children who received MR & bOPV in a mass campaign within a population-based cohort. The study was implemented by engaging VHT members and schoolteachers who were in direct contact with both the vaccine recipients as well as caretakers. Engaging them did not pose any difficulties during training or actual data collection on AEFIs that could affect the results. This is because the VHTs were more likely to have been exposed to basic data collection-related training, while the teachers were very interested and always keen on ensuring that learning and reporting was effective.

However, the study had limitations. First, AEFIs were identified by VHTs and schoolteachers with no specific clinical training in causality assessment. However, working with these community-based service providers mitigated the risk of vaccine hesitancy tendencies, since they were oriented on the benefits and potential risks associated with vaccines and could explain this information to their peers in the community. Secondly, the study team did not conduct thorough causality assessment between AEFIs and the administered antigen. This was solved by sharing all the AEFI reports and their classifications with NPC and the EPI for future investigation and causality assessment. Lastly, this study was not designed to capture rare events, but to demonstrate the utility of community-based structures in the rapid identification of AEFIs and communication of these events to the recipients, which counters the effects of rumours and sensational media reporting. Rare events need larger sample sizes and extended follow-up, which is costly but could be done using less costly electronic health records wherein data are captured over a long period of time [28,29].

## 5. Conclusions

Active vaccine pharmacovigilance identified more AEFIs during the national measles–rubella–polio (MR & bOPV) mass vaccination campaign than were reported through passive surveillance. The majority of the AEFIs were known events for these vaccines from clinical trials. AEFIs were nonserious and cleared within a day or two. Surveillance efforts that leverage on the use of community structures such as VHTs and schoolteachers complement the traditional passive reporting system, especially during mass vaccination campaigns. This not only increases the number of reports but builds public confidence in health system programs and mitigates vaccine hesitancy in LMICs. The active AEFI monitoring system provided extra information to national vaccine regulatory bodies. Countries need to conduct regular active reporting in order to obtain an accurate picture on overall AEFIs and to monitor trends and changes.

## Figures and Tables

**Figure 1 vaccines-09-01293-f001:**
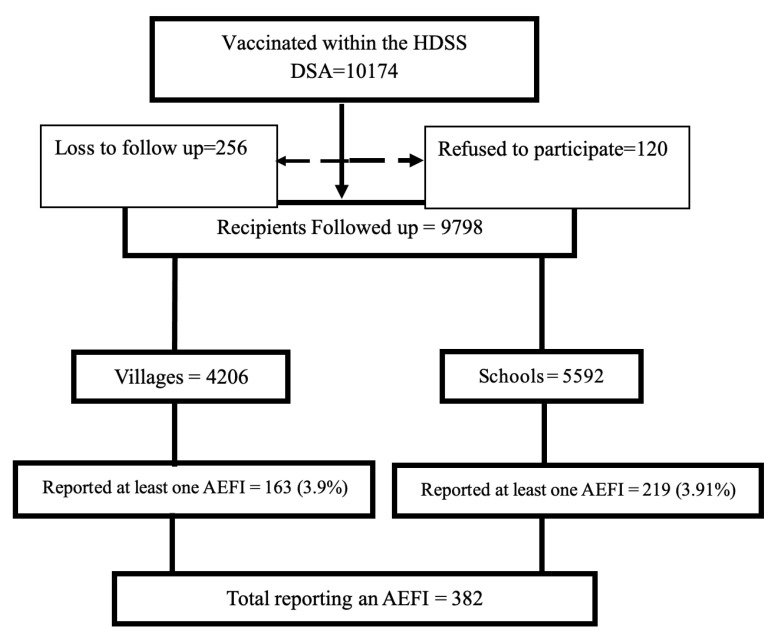
Flow diagram for the study. Legend—HDSS: health and demographic surveillance site; DSA: demographic surveillance area; AEFI: adverse event following immunization.

**Figure 2 vaccines-09-01293-f002:**
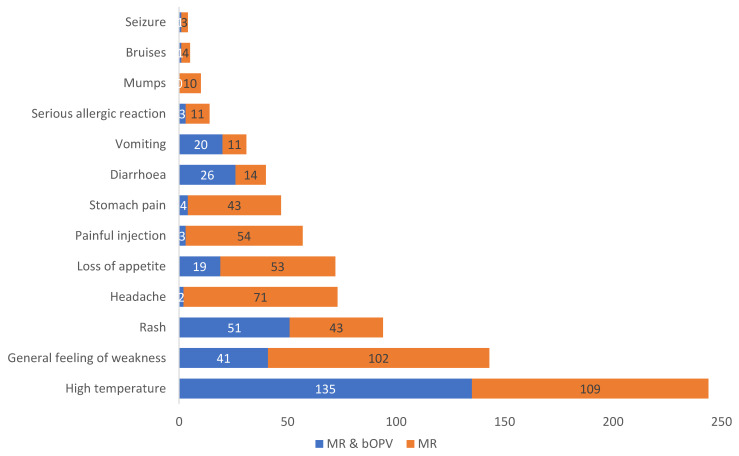
Number of AEFIs reported for individual vaccines.

**Figure 3 vaccines-09-01293-f003:**
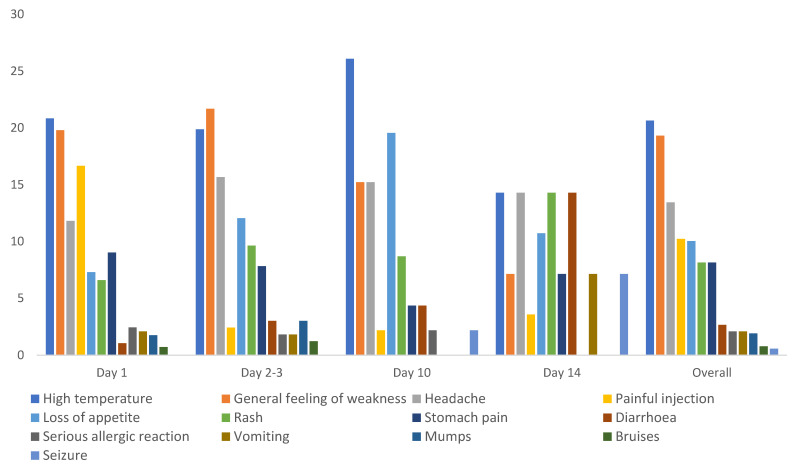
Number of AEFIs associated with mono MR and their time of occurrence.

**Figure 4 vaccines-09-01293-f004:**
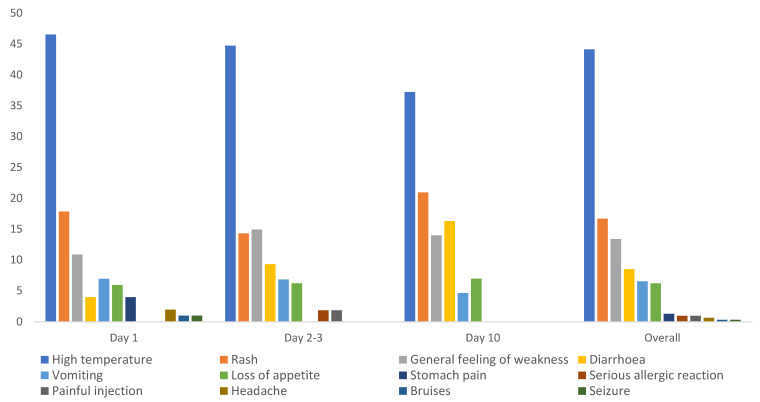
Number of AEFIs associated with combination of MR and polio by time of occurrence.

**Figure 5 vaccines-09-01293-f005:**
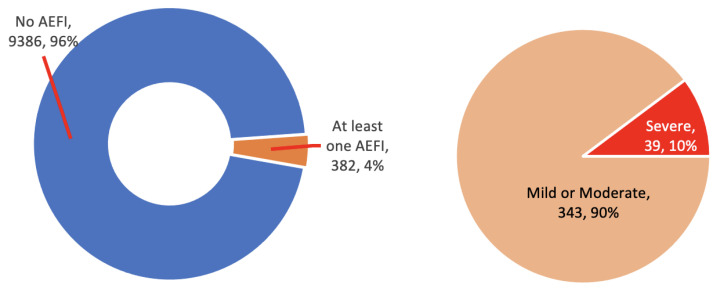
AEFI number and severity level as reported by the vaccine recipients or their caretakers.

**Table 1 vaccines-09-01293-t001:** Sociodemographic characteristics of the 382 (3.9%) children reporting any AEFI.

Variable	Category	Schools	VHTs *	Total (N = 382)
Gender	Male, n (%)	81 (49.4)	83 (50.6)	164 (42.9)
	Female, n (%)	138 (63.3)	80 (36.7)	218 (57.1)
Age	0–4 y, n (%)	19 (13.4)	123 (86.7)	142 (37.2)
	5–15 y, n (%)	200 (83.0)	40 (16.0)	240 (62.8)
**AEFIs reported**	**Reporting days**	**N = 302**	**N = 215**	**Total (N = 517)**
	Day 1, n (%)	165 (54.6)	60 (27.9)	225 (43.5)
	Day 2–3, n (%)	94 (31.1)	120 (55.8)	214 (41.4)
	Day 10, n (%)	29 (9.6)	28 (13.0)	57 (11.0)
	Day 14, n (%)	14 (4.6)	7 (3.3)	21 (4.1)

* VHTs–village health teams based in the community.

## Data Availability

The data presented in this study are available on request from the corresponding author (dan.kajungu@gmail.com). The data are publicly available after approval to share is acquired from multiple stakeholders including the district health office, the national regulatory authority, and the expanded program for immunization.

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
