# Peer review of "Using Population-Based Structures to Actively Monitor AEFIs during a Mass Immunization Campaign—A Case of Measles–Rubella and Polio Vaccines"

_vaccines, 2021, doi:10.3390/vaccines9111293_

Round 1

Reviewer 1 Report

It is not clear to me what you view as the value added to this paper.  If the purpose is to show that actively pursuing if AEFIs occur vs passive reporting events, then it seems to me that would be obvious at least to the point that active would also be equal or better than a passive approach.  If you were trying to report the incidence rate of AEFIs, the paper needs to be better strctured.  

The paper was not badly written, although you do have some repetition in the paper.   For example the section in lines 81-85 is repeated in lines109-112.  

I did not understand why double entry was necessary.  How was the data entered? I need to know a little more about the EPIDATA system and the recording process.  Was it text or a list entry?  Did previously immunized have any impact on those with AEs? 

I did not understand the sentence at line 242. 

The conclusions could have been made without the study.  What was the value of knowing AEs, particularly if not action is taken?

Author Response

Reviewer 1

It is not clear to me what you view as the value added to this paper.  If the purpose is to show that actively pursuing if AEFIs occur vs passive reporting events, then it seems to me that would be obvious at least to the point that active would also be equal or better than a passive approach.  If you were trying to report the incidence rate of AEFIs, the paper needs to be better strctured.  

Response:

Thank you for carefully reviewing our manuscript, and for the valuable comments and suggestions.

We agree that the paper needs to show value-added. We believe that active monitoring is necessary to improve AEFI reporting and complementing the conventional passive reporting systems, especially in mass campaigns. The WHO recommends active safety surveillance to improve reporting rates of AEFIs. We hope that this work will encourage colleagues in countries to modify their own systems and improve vaccine surveillance using the method described in this study. In addition, the results could also be used as a comparison and benchmark for future studies in places with similar settings.

We have gone ahead to revise the conclusion.

The paper was not badly written, although you do have some repetition in the paper. For example the section in lines 81-85 is repeated in lines109-112.  

Response:

Thank you. We have thoroughly reviewed the write-up and made the necessary modifications to improve the text.

I did not understand why double entry was necessary. How was the data entered? I need to know a little more about the EPIDATA system and the recording process. Was it text or a list entry? 

Response:

Thank you. Data on AEFIs were collected on paper questionnaires and needed to be entered into a specialized data entry program. Double data entry is always a good and recommended practice in research and was adopted to ensure consistency and minimize errors due to data entry, transcription. EPIDATA is a specialized data entry software that is commonly used by researchers for data entry because of the advantages it has over other data capture systems like Excel.

Did previously immunized have any impact on those with AEs? 

Thank you. This was not assessed as it is not an expected outcome- AE’s usually happen within the first few days of vaccination.

I did not understand the sentence at line 242. 

Response: Thank you for pointing this out. The sentence has been revised to provide clarity on the severity of adverse events in the revised article.

The conclusions could have been made without the study.  What was the value of knowing AEs, particularly if not action is taken?

Response:

Thank you for the comment. This work highlights the need to have an active follow-up of vaccine recipients during mass vaccination campaigns to be able to obtain the incidence of AEs using community-based structures. We have revised the conclusion to reflect this and reflect that the observed incidence can act as a reference point for future policy decisions regarding vaccines.

Reviewer 2 Report

The author conducted this study to determine adverse events following immunization (AEFI) with Measles and Rubella (MR), or MR in combination with the bivalent Oral Polio Vaccine (bOPV 1&3) (MR&OPV) during mass vaccination in Uganda. 

The method used are well described and answered directly to the objective of the study. 

The discussion and figures are excellent. I only have some minor language comments, as attached. 

Author Response

Reviewer 2

Comments and Suggestions for Authors

The author conducted this study to determine adverse events following immunization (AEFI) with Measles and Rubella (MR), or MR in combination with the bivalent Oral Polio Vaccine (bOPV 1&3) (MR&OPV) during mass vaccination in Uganda. 

The method used are well described and answered directly to the objective of the study. 

The discussion and figures are excellent. I only have some minor language comments, as attached. 

Response: Thank you for your very positive comments. We have thoroughly reviewed the manuscript and made edits and amendments to improve the language, grammar, and ease of readability of the new version.

Reviewer 3 Report

Authors evaluated effectiveness of using active approaches (including involvement of local health teams, community, and school teachers) in monitoring of the adverse events following immunization (AEFIs) in mass vaccination (~10,000 children) in Uganda which used Measles & Rubella (MR) and MR & Polio vaccine. They compared the outcomes of active approaches to passive reporting and found that active approach results in higher (and presumably more realistic in comparison to other studies) reports of AEFIs.

Study is timely and relevant and methodologically sound. I have no major concerns about the study and I can recommend it for publication assuming authors can resolve the listed minor comments.

Minor comments:

  • The AEFI incidence in this study is considerably lower (~4%) compared to other studies mentioned in the discussion (13.7% / 21.8%). Study would benefit from a short extension of the discussion of this discrepancy – e.g. did authors encounter considerable non-compliance among the monitored population? Or are the other studies done with very different setup / in different populations.
  • Could the authors comment if the study encountered non-compliance or other difficulties with training of AEFI monitoring teams / teachers and the Village Health Teams (VHTs)? A brief description of such difficulties (if any) in the discussion would be helpful for other researchers conducting similar studies / comparing the studies.
  • Figure 1: parts of schematic look incorrect, possibly due to PDF conversion glitch, please verify the plot is correct. Figure legend would benefit from explaining abbreviations in the legend for clarity / readability.
  • Manuscript has some (minor) typos and would benefit from extra round of proofreading.

Author Response

Reviewer 3

Comments and Suggestions for Authors

Authors evaluated effectiveness of using active approaches (including involvement of local health teams, community, and school teachers) in monitoring of the adverse events following immunization (AEFIs) in mass vaccination (~10,000 children) in Uganda which used Measles & Rubella (MR) and MR & Polio vaccine. They compared the outcomes of active approaches to passive reporting and found that active approach results in higher (and presumably more realistic in comparison to other studies) reports of AEFIs.

Study is timely and relevant and methodologically sound. I have no major concerns about the study and I can recommend it for publication assuming authors can resolve the listed minor comments.

 Thank you for describing our work as timely and relevant and for your valuable comments.

Minor comments:

  • The AEFI incidence in this study is considerably lower (~4%) compared to other studies mentioned in the discussion (13.7% / 21.8%). Study would benefit from a short extension of the discussion of this discrepancy – e.g. did authors encounter considerable non-compliance among the monitored population? Or are the other studies done with very different setup / in different populations.

Response: The other studies were conducted under routine immunization which was different from our study that was conducted during the mass immunisation campaign and follow-up lasted only 14 days. The other studies highlighted lasted a longer period of close to 3 years.

  • Could the authors comment if the study encountered non-compliance or other difficulties with training of AEFI monitoring teams / teachers and the Village Health Teams (VHTs)? A brief description of such difficulties (if any) in the discussion would be helpful for other researchers conducting similar studies / comparing the studies.

Response: The study team did not encounter any difficulties while training on AEFI monitoring teams/teachers and the Village Health Teams (VHTs) that could affect the results. Some of the VHTs have been exposed to research-related training. On the other hand, teachers were very interested and were key in ensuring that learning and reporting were effective. We have captured this in the discussion section (line 331-336)

  • Figure 1: parts of schematic look incorrect, possibly due to PDF conversion glitch, please verify the plot is correct. Figure legend would benefit from explaining abbreviations in the legend for clarity / readability.

Response: Thank you, this has been corrected. Legends have also been added to the figures to help in understanding.

  • Manuscript has some (minor) typos and would benefit from extra round of proofreading.

Response: Thank you. We have proofread the manuscript, thoroughly reviewed the spellings, typos, and grammatical errors. Edits and amendments have been included in the new version of the text and we hope to have now addressed all of these.

Round 2

Reviewer 1 Report

The authors have done an accerptable job in responding to my previous review.  A couple of minor comments:

_ What is mrtab command?

-There are 2 Figure 3s.  Need to be renumbered and reflected in the text.

- In the two Figure 3s, at first glance the key colors varible appear to relate to the time period above. Needs to make sure clearly not tied to Day.  Publisher may solve.

Author Response

The authors have done an accerptable job in responding to my previous review.  A couple of minor comments:

_ What is mrtab command?

Thank you so much for this comment. We have amended the sentence to capture what mrtab command is (line 151-152). The new sentence now reads thus: 'Since in some instances, multiple AEFIs are reported by a single child, the STATA mrtab command which tabulates multiple responses that are held as a set of indicator variables or as a set of polytomous response variables, was applied because of data-storage mode, that is, the indicator mode.'

-There are 2 Figure 3s.  Need to be renumbered and reflected in the text.

Thank you for point this out. We have changed the numbering and figure numbers are reflected in the text.

- In the two Figure 3s, at first glance the key colors varible appear to relate to the time period above. Needs to make sure clearly not tied to Day.  Publisher may solve.

Thank you, the legends of both figures show that the colors are for each reported AEFI